# Production of Value-Added Arabinofuranosyl Nucleotide Analogues from Nucleoside by an In Vitro Enzymatic Synthetic Biosystem

**DOI:** 10.3390/biom14111440

**Published:** 2024-11-13

**Authors:** Yuxue Liu, Xiaojing Zhang, Erchu Yang, Xiaobei Liu, Weiwei Su, Zhenyu Wang, Hailei Wang

**Affiliations:** Henan Engineering Research Center of Bioconversion Technology of Functional Microbes, College of Life Science, Henan Normal University, Xinxiang 453007, China; liuyuxue@htu.edu.cn (Y.L.); zhangxiaojing20229@stu.htu.edu.cn (X.Z.); yangechu@stu.htu.edu.cn (E.Y.); liuxiaobei2023@stu.htu.edu.cn (X.L.); 2404183082@stu.htu.edu.cn (W.S.)

**Keywords:** arabinoside, nucleoside, multi-enzyme cascade, biocatalysis

## Abstract

Arabinofuranosyl nucleotide analogue (arabinoside) and the derived compounds, a family of nucleoside analogues, exhibit diverse, typically biological activities and are widely used as antibacterial, antiviral, anti-inflammatory, and antitumor drugs in both clinical and preclinical trials. Despite their long and rich history in medicinal chemistry, the biosynthesis of arabinoside has only been sporadically designed and studied and has remained a challenging task. In this study, an in vitro synthetic enzymatic biosystem was designed and constructed for the production of arabinoside from low-cost nucleoside, based on a phosphorolysis -isomerization-dephosphorylation enzymatic cascade conversion routes. The enzymatic system achieves the biosynthesis of arabinoside by isomerizing the ribose part of nucleoside to arabinose. The reaction conditions affecting the yield of arabinoside were investigated and optimized, including meticulous enzyme selection, key enzyme dosage, the concentration of orthophosphate, and reaction time. Under the optimized conditions, we achieved the production of 0.12 mM of arabinofuranosylguanine from 0.5 mM of guanosine, representing 24% of the theoretical yield. Furthermore, this biosystem also demonstrated the capability to produce other arabinosides, such as vidarabine, spongouridine, and hypoxanthine arabinofuranoside from corresponding nucleosides. Overall, our biosynthesis approach provides a pathway for the biosynthesis of arabinoside.

## 1. Introduction

Because nucleosides play crucial roles in numerous critical biological processes, a number of nucleoside analogues show a broad spectrum of biological activities and have a long history in the field of medicinal chemistry [1,2]. Arabinose nucleosides, also known as arabinosylamines, form a substantial family of nucleoside analogues and are well known for their biological activities in clinical and preclinical trials. Arabinoside (Figure 1), such as vidarabine (Ara-A), cytarabine (Ara-C), spongouridine (Ara-U), arabinofuranosylguanine (Ara-G), nelarabine, clofarabine, fludarabine, and so on, enjoy substantial market demand, serving as staples in anticancer and antiviral treatments [3,4,5]. Notably, Ara-C and Ara-A are currently employed in anticancer treatment and against several viruses, primarily Herpes Simplex Virus infections, respectively [6,7,8,9]. Fludarabine is popular in the management of chronic lymphocytic leukemia [10].

Currently, chemical synthesis remains the primary approach for producing arabinosides. Most existing methodologies involve lengthy, multistep processes that require numerous protection and deprotection steps, alongside the use of organic solvents, which, while necessary, impose significant environmental burdens. Nelarabine, for instance, is routinely synthesized through a multi-step process starting from 2,3,5-*O*-benzyl-D-arabinofuranosyl chloride and 2-amino-6-chloropurine (Figure 2A) [11]. The key intermediate, chlorosugar 2,3,5-*O*-benzyl-D-arabinofuranosyl chloride, also serves as the precursor for the established chemical synthesis of Ara-A and must first be derived from D-ribose via a five-step process [11,12]. In contrast, the synthesis of Ara-U is relatively straightforward (Figure 2B) [13]. Ara-U can be readily prepared from uridine through the formation of a 2,2′-anhydronucleoside, followed by hydrolysis, yielding 68%. However, this method involves the use of diphenyl carbonate, an irritant, which presents safety concerns.

In this scenario, enzyme catalysis presents a convenient alternative. Utilizing either natural or engineered enzymes for organic synthesis, enzyme catalysis offers several advantages, including increased productivity, enhanced safety, tunability, and exceptional selectivity [14,15,16]. Substantial efforts have been devoted to developing biocatalytic methodologies for synthesizing arabinosides. Typically, nucleoside phosphorylases (NPs) and 2-deoxyribosyltransferases (NDTs) serve as powerful tools, employed individually or in combination, in both free and immobilized forms to catalyze transglycosylation reactions, either in vivo or in vitro [3,12,17,18,19,20]. In these processes, the substrate arabinoside undergoes catalysis to produce arabinose-1-phosphate (A1P) or arabinosyl-enzyme intermediate (Figure 3A). A new arabinoside is subsequently formed through the reaction of A1P with an acceptor nucleobase, often exhibiting high conversion rates. For example, Ara-C was successfully synthesized from Ara-A and cytosine with a conversion rate of 67.4% using a combination of purine phosphorylase 1 (PNP1) and uracil phosphorylase (UP) [17]. However, these methods primarily facilitate the enzymatic conversion between different arabinosides, necessitating the prior synthesis of substrate arabinosides through alternative means. Consequently, another strategy has emerged, focusing on the one-pot transformation of pentoses into nucleoside analogues in the presence of nucleobases, utilizing ribokinase (RK), phosphopentomutase (PPM), and NP (Figure 3B) [21]. While this multi-enzymatic system is effective, it is costly due to the high ATP consumption. The synthesis of arabinosides via this multi-enzyme system requires elevated concentrations of arabinose, up to 60 mM, which is 176 times higher than that of the nucleobases used. Furthermore, this approach is restricted to modified bases, making the incorporation of natural nucleobases into arabinose for arabinoside synthesis challenging.

In our previous study, we established an in vitro eight-enzymatic cascade for the biosynthesis of arabinosides from sucrose and nucleobase (Figure 3C) [22]. This system can produce various arabinosides and analogues by introducing different nucleobases; however, the conversion rate of nucleobases was low, with Ara-A yield from adenine at only 18.7%. Interestingly, alongside the target arabinosides, by-products of nucleosides were also generated, suggesting a potential side reaction involving the isomerization of arabinose phosphate or ribulose phosphate to ribose phosphate. Given that nucleosides are natural products, researchers have developed various cellular factories for efficient fermentation production [23,24,25,26,27]. This inspired us to design a multi-enzyme catalytic system to produce value-added arabinosides from nucleosides.

In this study, we constructed an in vitro synthetic enzymatic biosystem for the biosynthesis of arabinosides from low-cost nucleosides. Utilizing a phosphorolysis-isomerization-dephosphorylation cascade, the enzymatic system facilitates the conversion of the ribose component of nucleosides to arabinose. This biosystem incorporates four core enzymes, i.e., nucleoside phosphorylase (NP), phosphopentomutase (PPM), ribose 5-phosphate isomerase (RPI), and D-arabinose 5-phosphate isomerase (API). The biosystem were tested, analyzed, and systematically optimized, ultimately achieving a production yield of 24% for Ara-G from guanosine. This method not only enables the preparation of arabinosides but also paves the way for the de novo biosynthesis of arabinose-derived nucleoside analogues.

## 2. Materials and Methods

### 2.1. Materials and Chemicals

DNA polymerase and M5 SuperFast Seamless Assembly and Cloning mix were purchased from Mei5 Biotechnology (Beijing, China). Glucose 1,6-biophosphate, nucleobase standards (adenine, guanine, uracil, hypoxanthine), Ara-A, Ara-G, Ara-U, hypoxanthine arabinofuranoside (Ara-I), Isopropyl-β-thiogalactoside (IPTG), and kanamycin were purchased from Shanghai Biotech (Shanghai, China) or Shyuanye (Shanghai, China). All other chemical reagents used were analytical grade and commercially available.

### 2.2. Strains and Plasmids

The bacterial strains and plasmids utilized in this study are listed in Appendix A. *Escherichia coli* strain DH5α was used for plasmid construction, while BL21 (DE3) was utilized for recombinant protein overexpression. An LB (Luria–Bertani) culture medium (10 g/L of tryptone, 5 g/L of yeast 36 extract, and 10 g/L of NaCl) was employed for strain growth and protein expression.

### 2.3. Construction of Plasmids and Strains

Ten plasmids were utilized in this study: pET28a-*Ec*PNP encoding PNP1 (*Ec*PNP) from *E. coli*, pET28a-*Ec*XP encoding PNP (*Ec*XP) from *E. coli*, pET28a-*Ec*UP encoding UP (*Ec*UP) from *E. coli*, pET28a-*Ec*TP encoding thymidine phosphorylase (*Ec*TP) from *E. coli*, pET28a-*Kl*PNP encoding PNP (*Kl*PNP) from *Klebsiella*, pET28a-*Aa*PNP encoding PNP (*Aa*PNP) from *Alicyclobacillus acidoterrestris* (*deoD*), pET28a-*Bs*PNP encoding PNP (*Bs*PNP) from *Bacillus subtilis*, pET28a-*Bl*PNP encoding PNP (*Bl*PNP) from *Bacillus licheniformis*, pET28a-*Bc*PPM encoding phosphopentomutase (*Bc*PPM) from *Bacillus cereus*, and pET28a-*Hs*RK encoding ribokinase isoform 1 (*Hs*RK) from *Homo sapiens*. These plasmids were obtained from our previous research and laboratory collection [22]. The gene *rpiA* encoding ribose-5-phosphate isomerase A (*Ec*RpiA) from *E. coli* was cloned into pET28a between the *Nco I* and *Xho I* site, in-frame with the C-terminal his-tag, to yield the plasmid pET28a-rpiA.

### 2.4. Protein Expression and Purification

All plasmids were transferred into *E. coli* BL21 (DE3) for protein expression. Single colony containing the desired plasmid was cultivated in an LB broth medium at 37 °C with shaking at 180 rpm. When necessary, 50 μg/mL of kanamycin was supplemented. Cells were overexpressed by growing them at 37 °C in the LB broth supplemented with 50 μg/mL of kanamycin until the absorbance reached 0.4 to 0.6 at A_600_. The induction of protein expression was then achieved by shifting the temperature to 25 °C and adding 0.5 mM of IPTG, with cultivation continued for 12 h. Cells were harvested, washed, and resuspended in a native binding buffer (50 mM of NaH_2_PO_4_, 500 mM of NaCl, pH 8.0) before being disrupted by sonication. Cell debris was removed by centrifugation at 12,000× *g* for 40 min at 4 °C. The target proteins in the supernatant were purified using a Ni-NTA purification kit (Invitrogen, Carlsbad, CA, USA). Purified proteins were stored at −80 °C in 50 mM of Tris-Cl (pH 7.5) containing 10% glycerol (*v*/*v*). Protein purity was assessed by SDS-PAGE, and concentrations were determined using a Bradford protein assay kit (Bio-Rad, Hercules, CA, USA).

### 2.5. Enzymatic Activity Assays

The activities of NPs towards arabinosides, including Ara-A, Ara-C, Ara-U, and Ara-I, were determined in a 50 mM potassium phosphate buffer (pH 7.5) containing 1 mM of arabinoside and 1 mg/mL of NP. Reactions were initiated by adding the enzyme solution and incubated at 45 °C for 45 min. Nucleobases and arabinosides were quantified by high-performance liquid chromatography (HPLC) as described below. The conversion rate was calculated using the following Equation (1):(1)$$\mathrm{Conversion}\;\mathrm{rate}\;(\%)=[\mathrm{Nucleobase}]/\left[\mathrm{Nucleobase}+\mathrm{Residual}\;\mathrm{nucleobase}\right]\;\times \;100\%$$

The phosphorolytic activities of PNPs towards Ara-G or guanosine were determined in a 50 mM potassium phosphate buffer at pH 7.5 containing 1 mM of Ara-G or guanosine and 0.1 mg/mL of PNP. The reaction was initiated by adding an enzyme solution and was incubated at 40 °C for 15 min, then quenched by heating. Guanine was quantified by HPLC. The unit of PNP activity for the phosphorolysis of Ara-G was defined as the amount of enzyme that released 1 μmol of guanine per minute.

The dephosphorolytic activities of PNPs for the synthesis of Ara-G were determined as follows: Firstly, D-arabinose 1-phosphate was accumulated in a reaction mixture containing 2 mM of Ara-U and 1 mg/mL of *Ec*UP in a 50 mM potassium phosphate buffer at pH 7.5. The reaction mixtures were incubated at 45 °C for 15 min and quenched by heating. Protein was removed through centrifugation at 12,000× *g* for 2 min at 4 °C. Next, an aliquot with a volume of 100 μL was transferred to a substrate solution containing 2 mM of guanine and 0.4 mg/mL of PNP. This reaction was incubated at 45 °C for 15 min and quenched by heating. Ara-G production was determined by HPLC, and the unit of PNP activity for Ara-G synthesis was defined as the amount of enzyme that produced 1 μmol of Ara-G per minute.

The activity of PPM from *Bacillus cereus* (*Bc*PPM) was assessed by coupling it with PNP to catalyze the synthesis of Ara-A as follows. D-arabinose 5-phosphate was biosynthesized in a reaction mixture (0.25 mL, 50 mM of Tris-HCl, pH 8.0) containing 4 mM of ATP, 5 mM of D-arabinose, 0.6 mM of MgCl_2_, 6 mM of KCl, and 0.25 mg of *Hs*RK. This mixture was incubated at 45 °C for 20 min and quenched by heating. Proteins were removed through centrifugation at 12,000× *g* for 2 min at 4 °C. Subsequently, an aliquot with a volume of 100 μL was transferred to 100 μL of a substrate solution containing 2 mM of adenine, 1.2 mM of MnCl_2_, 1 mM of glucose-1,6-biophosphate, 0.2 mg of *Kl*PNP and 0.2 mg of *Bc*PPM. The reaction was incubated at 45 °C for 45 min and quenched by heating. Ara-A production was quantified by HPLC, and the unit of *Bc*PPM activity was defined as the amount of enzyme that produced 1 μmol of Ara-A per minute.

### 2.6. Production of Arabinosides from Nucleoside

To evaluate the feasibility of Ara-G production from guanosine, reactions were conducted in the reaction mixture containing 0.5 mM of guanosine, 6 μM of *Ec*RpiA, 6 μM of *Ec*API, 1 U/L of *Bc*PPM, and 4.5 U/L of *Kl*PNP in a 50 mM Tris-HCl buffer (pH 8.0) at 45 °C for 1 h. *Bc*PPM was pre-activated at a concentration 10-fold higher than that used in the reactions by incubation for 10 min at room temperature in a 50 mM Tris-HCl buffer (pH 8.0), 0.1 mM of MnCl_2_ and 0.01 mM of glucose-1,6-biophosphate, before being added to the reaction mixture. The production of other arabinosides via the in vitro multi-enzymatic cascade was performed under similar conditions, using 0.5 mM of the corresponding nucleoside as the substrate.

The effect of orthophosphate on the Ara-G production from guanosine was examined by varying the concentration of K_2_HPO_4_, from 0.05 mM to 1 mM. For studying the impact of individual concentrates on Ara-G production, the concentration of *Ec*RpiA, *Ec*API, *Bc*PPM or *Kl*PNP were adjusted. The concentration ranges for *Ec*API and *Ec*RpiA were set at 6 μM, 9 μM, 15 μM, 18 μM, and 21 μM. The loading amounts of *Bc*PPM were varied from 1 U/L to 3.2 U/L and *Kl*PNP from 4.5 U/L to 16 U/L. Nucleobases, nucleoside, and arabinosides in reaction samples were quantified by HPLC.

### 2.7. Analytical Methods

Nucleobases and arabinosides were quantified at 259 nm using an HPLC system equipped with an SPD 20A DAD detector (Shimadzu, Beijing, China). A reversed-phase C18 column (250 mm × 4.6 mm, 5 μm) was utilized, with a mobile phase of methanol/H_2_O (15:85, *v*/*v*) at a flow rate of 1.0 mL/min. All assays were performed in triplicate.

The Gibbs free energy (ΔG°) was analyzed using the eQuilibrator website (https://equilibrator.weizmann.ac.il/ (accessed on 12 September 2024)) at pH 7.5, and the ionic strength of 0.25 M. Standard concentrations for reactants and products was assumed to be 1 M in the computation.

## 3. Results and Discussion

### 3.1. In Vitro Pathway Design for Arabinoside Production from Nucleoside

The configuration of the hydroxyl group at the C2 position in arabinosides is inverted compared to that in nucleosides. In vivo, isomerization can be catalyzed by isomerase. For example, the interconversion between aldopentose/aldohexose phosphate and ketopentose/ketohexose phosphate is catalyzed by phosphate sugar isomerases, which are widespread in various organisms and play a crucial role in microbial sugar metabolism [28]. Phosphate sugar isomerases are also employed in the industrial production of valuable rare sugars. Although the potential for isomerization between arabinosides and nucleosides via oxidoreductases has been proposed, it remains uncharacterized [29]. We hypothesize that the isomerization between arabinosides and nucleosides can be effectively achieved through a phosphorolysis–isomerization–dephosphorylation enzymatic cascade conversion.

An in vitro synthetic enzymatic cascade reaction was designed and implemented for the transformation of nucleoside to arabinoside (Figure 1A). This biosystem incorporates four core enzymes, facilitating the following reactions: (1) phosphorolysis of the nucleoside to ribose 1-phosphate (R1P) and the corresponding nucleobase, catalyzed by nucleoside phosphorylase (NP) in the presence of orthophosphate; (2) the conversion of R1P to ribose 5-phosphate (R5P) catalyzed by phosphopentomutase (PPM); (3) the conversion of R5P to ribulose 5-phosphate (Ru5P) catalyzed by ribose 5-phosphate isomerase (RPI); (4) the conversion of Ru5P to arabinose 5-phosphate (A5P) catalyzed by D-arabinose 5-phosphate isomerase (API); (5) the conversion of A5P to arabinose 1-phosphate (A1P) catalyzed by PPM; and (6) the dephosphorylation of A1P and nucleobase to produce arabinoside and orthophosphate, catalyzed by NP. In this four-enzyme catalytic system, NP and PPM catalyze two reactions each. The nucleobase and orthophosphate can be recycled from nucleoside to arabinoside, while orthophosphate can be reused between reactions (1) and (6) within a single vessel.

The Gibbs free energy change (ΔG°) of the overall route from nucleoside to arabinoside was analyzed to be around zero (Figure 1B,C). In contrast, the ΔG° of the overall pathway from R1P to A1P was analyzed to be −2.3 ± 2.4 kJ mol^−1^ (Appendix A), and it is thermodynamic favourable to the synthesis of purine nucleoside and purine arabinoside from pentose 1-phosphate and nucleobase. This computation suggests that the thermodynamics of the conversion process do not significantly enhance the yield of arabinosides from nucleosides.

### 3.2. Enzyme Selection of NP

NP is crucial for arabinoside production in this multi-enzymatic cascade, as it reversibly catalyzes the phosphorolysis of nucleosides and the biosynthesis of arabinosides from A1P and nucleobase. NPs are classified based on their substrates and quaternary structures into purine nucleoside phosphorylase (PNP, EC 2.4.2.1), pyrimidine nucleoside phosphorylase (PyNP, EC 2.4.2.2), uridine phosphorylase (UP, EC 2.4.2.3), and thymidine phosphorylase (TP, EC 2.4.2.4) [30]. Equilibrium constants across these enzymes are shown in Table 1. Due to their broad substrate specificity, several enzymatic nucleoside analogues have been biosynthesized using NP as a biocatalyst [19,30]. Notably, PNPs sourced from *Aeromonas hydrophila*, *Thermus thermophilus HB27*, and *E. coli*, as well as UP from *Clostridium perfringens*, have been employed for synthesizing arabinosides or base-modified arabinosides [20,21,31].

In this study, we selected eight different NPs sourced from various microorganisms as candidates: PNP from *Klebsiella* (*Kl*PNP), PNP from *E. coli* (*Ec*PNP and *Ec*XP), UP from *E. coli* (*Ec*UP), PNP from *Alicyclobacillus acidoterrestris* (*Aa*PNP), TP from *E. coli* (*Ec*TP), PNP from *Bacillus subtilis* (*Bs*PNP), and PNP from *Bacillus licheniformis* (*Bl*PNP). These eight NPs were overexpressed in *E. coli* BL21 (DE3) and purified to homogeneity (Appendix A).

The NPs exhibited significant variations in their abilities to phosphorylate different arabinosides (Figure 2A). Specifically, *Ec*PNP, *Ec*UP, and *Ec*TP displayed high activity exclusively towards Ara-U, while *Ec*XP showed high activity solely towards Ara-G. In contrast, *Aa*PNP and *Kl*PNP were highly active towards Ara-A, Ara-G, and Ara-I, whereas *Bs*PNP and *Bl*PNP demonstrated higher activity towards Ara-A but lower activity towards Ara-G and Ara-I. We subsequently selected four PNPs with differing hydrolytic capacities towards Ara-G to assess their phosphorolysis and dephosphorylation activity with guanosine or Ara-G. As illustrated in Figure 2B, *Aa*PNP, *Kl*PNP, *Bs*PNP and *Bl*PNP exhibited relatively high phosphorolysis activity towards guanosine, with values of 321 U/g, 222 U/g, 323 U/g, and 292 U/g, respectively. The dephosphorylation activity of *Aa*PNP and *Kl*PNP for guanosine were 131% and 195% of their phosphorolysis activities, respectively, while *Bs*PNP and *Bl*PNP exhibited very low dephosphorylation activities, accounting for only 17% and 24% of their phosphorolysis activities. The phosphorolysis activities of these enzymes towards Ara-G were low, yielding values of 123 U/g, 147 U/g, 28.4 U/g, and 38 U/g for *Aa*PNP, *Kl*PNP, *Bs*PNP, and *Bl*PNP, respectively, with minimal dephosphorylation activity observed. No dephosphorylation activity of *Bs*PNP towards Ara-G was detected, while the others achieved only 8% of their phosphorolysis activity, with the highest dephosphorylation activity of *Aa*PNP being a mere 10 U/g. These findings align with our previous studies, which indicated that the phosphorolysis activity of PNPs towards arabinosides significantly exceeds their dephosphorylation activity [22].

In our multistep biocatalytic system, the optimal PNP should exhibit high phosphorolysis activity towards nucleoside and substantial dephosphorylation activity towards arabinoside. Conversely, high dephosphorylation activity towards nucleoside and high phosphorolysis activity towards arabinoside are unfavourable for arabinoside production. Based on these results, the selected PNP does not contain an enzyme that is optimal for the given purpose. For subsequent proof-of-concept, *Kl*PNP was randomly chosen to catalyze the synthesis of Ara-G.

### 3.3. Enzyme Selection of Phosphate Pentose Isomerase

In this in vitro biosystem, three phosphate pentose isomerases, including phosphopentomutase (PPM), D-arabinose 5-phosphate isomerase (API), and ribose-5-phosphate isomerase (RPI), catalyze the cascade reaction converting R1P to A1P.

PPM interconverts R1P and R5P, as well as A1P and A5P, within our artificial multi-enzymatic cascade. PPM from *Bacillus cereus* (*Bc*PPM) has been well characterized and successfully utilized in the in vitro production of islatravir, making it a natural candidate for our biosystem. The catalytic activity of *Bc*PPM was measured, yielding 55.2 U/g for the conversion of R5P to R1P and 14.6 U/g for the conversion of A5P to A1P. The relatively low activity of A5P to A1P may present a rate-limiting step in our catalytic system.

D-arabinose 5-phosphate isomerase (API) specifically catalyzes the interconversion between A5P and Ru5P. In *E. coli*, three traditional APIs are known, including the following: KdsD, GutQ, and KpsF [32]. The equilibrium for catalysis by these APIs favours A5P, with an equilibrium constant (*K_eq_*) between 0.48 and 0.5 ([Ru5P]/[A5P]) (Table 1). While they exhibit similar *K*_m_ values, KdsD has the highest *k*_cat_/*K*_m_ [32,33]. Therefore, *E. coli* KdsD (*Ec*KdsD) was selected for the subsequent arabinoside production.

Ribose-5-phosphate isomerase (RPI), while including type A (RpiA) and type B (RpiB), is widely distributed in organisms and catalyzes the isomerization between D-ribulose 5-phosphate and D-ribose 5-phosphate [34]. RpiA, identified in *E. coli* (*Ec*RpiA), accounts for approximately 99% of the total RPI activity in this organism [35]. Consequently, *Ec*RpiA was selected for our subsequent study.

### 3.4. Validation of One-Pot Biosynthesis of Ara-G

Before conducting the proof-of-concept experiment, Ara-G was selected as the target for biosynthesis. Given our previous construction of an engineered *E. coli* strain capable of accumulating guanosine [36], this in vitro pathway could potentially be adapted for in vivo biosynthesis if proven cost-effective for converting guanosine into Ara-G.

For the proof-of-concept experiment, all enzymes were purified to homogeneity (Appendix A). The reaction for the production of Ara-G was performed. As shown in Figure 3A, the HPLC peak corresponding to Ara-G, which was further identified by high performance liquid chromatography-mass spectrometry, was detected in all reactions containing *Kl*PNP and *Bc*PPM, indicating the feasibility of synthesizing Ara-G from guanosine using this in vitro biosystem. The concentration of Ara-G was 48.6 μM when catalyzed by the four-enzymatic cascade, while it was 35.5, 16.4, and 13.4 μM with the absence of RPI, API, or both RPI and API, respectively (Figure 3B). However, in the absence of either *Kl*PNP or *Bc*PPM, guanosine was not converted to Ara-G. Although RPI and API are not essential, their presence significantly increases Ara-G yield. These results demonstrate the viability of our proposed enzymatic cascade for the production of Ara-G from guanosine.

Based on these results, it is evident that R5P can be converted to A5P even in the absence of *Ec*RpiA and *Ec*KdsD.

Despite R5P and Ru5P being natural products with theoretical concentrations of 787 μM and 112 μM, respectively, in *E. coli* [37], the lack of arabinoside accumulation in most microorganisms is noteworthy. Our previous analysis of nucleoside phosphorylase (PNP) enzyme activity suggests that this phenomenon is primarily due to the significantly higher hydrolytic activity of PNP compared to its synthetic activity.

### 3.5. Reaction Conditions and Production Capacity of Ara-G

The yield of Ara-G in the proof-of-concept experiment was significantly lower than the theoretical maximum. Therefore, we investigated the primary reaction conditions affecting the biosynthesis of arabinoside, including orthophosphate concentration, enzyme loading amounts, and reaction time.

In our four-enzyme catalytic system, orthophosphate influences two catalytic reactions: the NP-catalyzed reversible phosphorolysis reaction and the PPM-catalyzed pentose phosphate isomerization. High concentrations of orthophosphate can inhibit PPM activity [38]. Additionally, NPs catalyze the reversible phosphorolysis of arabinoside in the presence of orthophosphate, which may promote phosphorolysis reactions of both nucleoside and arabinoside, thereby hindering arabinoside synthesis. Consequently, the orthophosphate concentration in the system is critical for the production rate of arabinosides.

We optimized the orthophosphate concentration in the one-pot multi-enzyme catalyzed biosystem across a range of 0.05 to 1 mM. As shown in Table 2, the production of Ara-G and the residual guanosine decreased with increased orthophosphate. When the concentration of orthophosphate is 0.05 mM, the yield of Ara-G reaches its peak. At this point, orthophosphate was recycled at least 2.3 times. In the subsequent verification experiment, the orthophosphate was added at 0.05 mM. At various time points, samples of the reaction solution were subjected to HPLC for detection. The results in Figure 4A indicate that the yield of Ara-G initially increased to 0.1 mM before declining with the extended reaction time. Concurrently, the concentration of guanosine gradually decreased, while the level of the intermediate metabolite guanine increased. It suggests that the phosphorolysis of nucleosides gradually became more dominant than their dephosphorylation during the reaction.

The effect of enzyme quantity on the yield of Ara-G was investigated one by one under the optimum orthophosphate concentration (Figure 4B). When the initial concentration of guanosine was 0.5 mM, the concentration of *Kl*PNP was changed from 6 to 21 μM (1 to 3.5 U/L) while the other three enzyme loadings were 6 μM each. The productivity of Ara-G rapidly increased from 15 μM/h to 111 μM/h when the concentration of *Kl*PNP was increased from 6 to 21 μM. In contrast, the loading amounts of PPM, API, and RpiA within the detected range had little or no impact on the production yield of Ara-G. Overall, under the optimized conditions, we achieved 0.12 mM of Ara-G from 0.5 mM of guanosine, representing 24% of the theoretical yield.

### 3.6. Production of Other Arabinosides Through the In Vitro Multi-Enzymatic Cascade

To expand the application of the in vitro multi-enzymatic cascade, we introduced different NPs or nucleosides to produce additional arabinosides. Based on the NP enzyme activity analysis mentioned above, we utilized *Kl*PNP to produce Ara-A and Ara-I, and *Ec*UP to produce Ara-U. As depicted in Figure 5, the introduction of 0.5 mM of adenosine or inosine to replace guanosine in the one-pot reaction system mentioned above resulted in the production of Ara-A and Ara-I at a concentration of 182 μM and 6.2 μM, respectively. Additionally, by substituting *Kl*PNP with *Ec*UP, 13 μM of Ara-U was produced from uridine, using the in vitro multi-enzymatic cascade. Unfortunately, the yield of Ara-I and Ara-U were very low compared to Ara-G and Ara-A, which was related to the synthetic activity of *Kl*PNP and the thermodynamic tendency of UP to catalyze the hydrolysis of nucleosides [39]. To improve the yield, it may be necessary to optimize the phosphate concentration and other conditions for the multi-enzyme reaction of Ara-I and Ara-U synthesis. Further investigation into these parameters and selecting and engineering enzymes will be essential for enhancing the efficiency and applicability of the multi-enzymatic cascade for the production of a wider range of arabinosides.

## 4. Conclusions

In this study, we designed a multi-enzymatic cascade for the production of arabinoside from low-cost nucleoside. A key advantage of this system is its ability to recycle orthophosphate and nucleobases, facilitating the conversion of nucleosides into arabinosides. After optimizing the biosynthetic conditions, we successfully produced 0.12 mM of Ara-G from 0.5 mM of guanosine, achieving 24% of the theoretical yield. Additionally, this system can synthesize other arabinosides, including Ara-A, Ara-I, and Ara-U, by incorporating different nucleoside phosphorylases.

## Data Availability

The data presented in this study are available on request from the corresponding author due to the large amount of raw data and the limitation of professional software.

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
