# Peer review of "Production of Value-Added Arabinofuranosyl Nucleotide Analogues from Nucleoside by an In Vitro Enzymatic Synthetic Biosystem"

_biomolecules, 2024, doi:10.3390/biom14111440_

Round 1
Reviewer 1 Report
Comments and Suggestions for Authors
In their study, "Production of value-added arabinoside from nucleoside by an in vitro enzymatic synthetic biosystem," Liu et al. present a novel approach for synthesizing value-added arabinoside derivatives from nucleoside substrates. The research addresses the growing demand for efficient and sustainable production of arabinoside-based compounds, which hold significant therapeutic and industrial potential. By leveraging an in vitro enzymatic biosystem, the authors demonstrate a method that bypasses the limitations of traditional chemical synthesis, offering a cleaner and more specific pathway for arabinoside production. This enzymatic approach optimizes reaction conditions and enzymatic pathways to enhance yield and reduce by-products, marking a promising advancement in biocatalytic synthesis techniques.
The study and its incentive are based-off a good scientific methodology and follows-up previously published work.
This publication is recommendable for publication however after substantial modifications.
Here´s a list below:
· Regarding the yield of 24% which is apparently 3x lower than the state-of-the art chemical synthesis using diphenyl carbonate. It would be good, from the very abstract, to precise the advantage of the method developed in the study. (i.e. milder conditions, no hazardous chemicals etc). This could increase interest from potential readers that discover biocatalysis.
· I noticed the term inorganic phosphate is used throughout the manuscript. Given that it can refer to several phosphate species (e.g., orthophosphate, pyrophosphate), it might be helpful to specify which form is intended for clarity. Specifying the exact species (e.g., orthophosphate if that's the case) will help readers better understand the precise chemical context and avoid any ambiguity regarding the role of phosphate in the reaction.
· In addition, it would seem logical to examinate different inorganic phosphate forms such as polyphosphate (e.g., sodium tripolyphosphate or polyphosphoric acid) in your system. Polyphosphates can provide multiple phosphate groups, which might enhance reaction efficiency or influence enzyme activity. Investigating the effects of polyphosphate could offer further insights into optimizing the biosynthetic pathway and improving product yields.
· I strongly advise including SDS-PAGE results to provide a visual representation of enzyme purity, along with e.g. grayscale analysis for quantification. This information is essential for validating the purity of the enzyme preparation. Additionally, sharing the gene sequence alongside the plasmid details used for overexpression would enhance the reproducibility of your work and allow other researchers to better understand and utilize your findings.
· The discussion (line 220-223) on nucleobase and inorganic phosphate recycling lacks quantitative data on the efficiency of these processes. Including a table to summarize the recycling rates would enhance clarity and provide a more comprehensive view of the system's performance. This could help illustrate the effectiveness of the recycling strategies employed and allow readers to grasp the practical implications of your findings more readily.
· I recommend including a table to compare the equilibrium constants across different enzymes discussed in the manuscript. This would allow for a clearer understanding of their relative efficiencies. Additionally, presenting kinetic data, such as reaction rates and Michaelis-Menten parameters, would provide further insights into enzyme performance and help contextualize your findings within the broader field. Such data would significantly enhance the depth of your analysis and support your conclusions.
· The paragraph spanning lines 309 to 317 contains descriptions of methods but is in the discussion section. It may be more appropriate to move this content to the methods section to maintain clarity and ensure that the discussion focuses on interpreting the results. This adjustment would help streamline the manuscript and enhance the overall organization.
· While the enzymatic system is well thought out, I noticed that there is no mention of how the product of interest will be recovered from the mixture. Although I understand that developing a detailed downstream processing (DSP) methodology may not be necessary, it would be beneficial to at least outline potential purification strategies for the target compound(s). Given the complexity of the one-pot system, which contains enzyme buffer and other components, recovery could present significant challenges that might influence the overall feasibility of your work. Including this information would enhance the manuscript by providing a more complete picture of the practical application.
· The metrics discussed in the study should include space-time yields, as this measurement is critical for evaluating the efficiency of the enzymatic system. Comparing space-time yields with those from other studies or enzyme systems would provide valuable context for your findings and help demonstrate the potential advantages of your approach. This addition could strengthen the overall impact of the manuscript by highlighting its contributions to the field.
· Assessing how the enzyme activity declines under the given reaction conditions would provide valuable information about the system's stability and help identify factors that may contribute to enzyme degradation. This data could be crucial for understanding the long-term viability of the enzymatic process and optimizing reaction conditions to enhance enzyme longevity.
Comments on the Quality of English Language· I noticed that the term "thermodynamic dynamics" appears in line 228. It’s a bit redundant since thermodynamics already encompasses dynamic processes related to energy changes. Simplifying this to just "thermodynamics" would enhance clarity. Additionally, I've come across similar instances throughout the text. It might be beneficial to have the paper reviewed by a native English speaker or a professional editing service to improve the overall language quality. This could help strengthen the manuscript and ensure your findings are communicated as effectively as possible.
Author Response
Comment 1. I noticed the term inorganic phosphate is used throughout the manuscript. Given that it can refer to several phosphate species (e.g., orthophosphate, pyrophosphate), it might be helpful to specify which form is intended for clarity. Specifying the exact species (e.g., orthophosphate if that's the case) will help readers better understand the precise chemical context and avoid any ambiguity regarding the role of phosphate in the reaction.
Response 1: We appreciate your insightful comment. In this study, the term "inorganic phosphate" refers specifically to orthophosphate, and we have revised the manuscript to clarify this by replacing "inorganic phosphate" with "orthophosphate" throughout. This should help eliminate any potential ambiguity and provide a clearer understanding of the chemical context.
Comment 2. In addition, it would seem logical to examinate different inorganic phosphate forms such as polyphosphate (e.g., sodium tripolyphosphate or polyphosphoric acid) in your system. Polyphosphates can provide multiple phosphate groups, which might enhance reaction efficiency or influence enzyme activity. Investigating the effects of polyphosphate could offer further insights into optimizing the biosynthetic pathway and improving product yields.
Response 2: We appreciate your thoughtful suggestion. In our reaction system, orthophosphate is recycled, and high concentrations of orthophosphate have been shown to inhibit PPM activity while accelerating the hydrolysis of the target product, Ara-G. For this reason, we did not consider it necessary to introduce polyphosphates to provide additional phosphate groups. However, we did test sodium hexametaphosphate in the system. The yield of Ara-G increased with the concentration of sodium hexametaphosphate in the range of 0.01 to 0.1 mM, reaching a maximum of 5.7%. This yield was still significantly lower than the conversion rate achieved using orthophosphate.
Comment 3. I strongly advise including SDS-PAGE results to provide a visual representation of enzyme purity, along with e.g. grayscale analysis for quantification. This information is essential for validating the purity of the enzyme preparation. Additionally, sharing the gene sequence alongside the plasmid details used for overexpression would enhance the reproducibility of your work and allow other researchers to better understand and utilize your findings.
Response 3: Thank you for your valuable suggestion. SDS-PAGE results have been included in the revised supporting information to provide a visual representation of enzyme purity. Additionally, the gene sequence and plasmid details used for overexpression have been added or appropriately cited in the revised manuscript, which will help improve the reproducibility of our work and allow for better understanding and utilization of our findings by other researchers.
Comment 4. The discussion (line 220-223) on nucleobase and inorganic phosphate recycling lacks quantitative data on the efficiency of these processes. Including a table to summarize the recycling rates would enhance clarity and provide a more comprehensive view of the system's performance. This could help illustrate the effectiveness of the recycling strategies employed and allow readers to grasp the practical implications of your findings more readily.
Response 4: Thank you for your constructive comment. The discussion on nucleobase and inorganic phosphate recycling (lines 220-223) is still in the theoretical phase, and the summary of recycling efficiency is provided in Section 3.5. The recycling efficiency of orthophosphate was calculated based on the concentration of orthophosphate used and the yield of Ara-G. Specifically, when the concentration of orthophosphate was 0.05 mM, it was recycled at least 2.3 times. We have considered your suggestion to include a table summarizing the recycling rates and will incorporate this in the revised manuscript for better clarity and to highlight the practical implications of our findings.
Comment 5. I recommend including a table to compare the equilibrium constants across different enzymes discussed in the manuscript. This would allow for a clearer understanding of their relative efficiencies. Additionally, presenting kinetic data, such as reaction rates and Michaelis-Menten parameters, would provide further insights into enzyme performance and help contextualize your findings within the broader field. Such data would significantly enhance the depth of your analysis and support your conclusions.
Response 5: We appreciate your thoughtful suggestion. A table (Table 1) comparing the equilibrium constants of the different enzymes discussed in the manuscript has been included in the revised version. Additionally, we will consider incorporating kinetic data, such as reaction rates and Michaelis-Menten parameters, in future studies to provide deeper insights into enzyme performance and further contextualize our findings within the broader field.
Comment 6. The paragraph spanning lines 309 to 317 contains descriptions of methods but is in the discussion section. It may be more appropriate to move this content to the methods section to maintain clarity and ensure that the discussion focuses on interpreting the results. This adjustment would help streamline the manuscript and enhance the overall organization.
Response 6: Thank you for your helpful comments. The descriptions of methods in lines 309 to 317 have been moved to the Methods section in the revised manuscript. This adjustment helps to streamline the content and ensures that the Discussion section remains focused on interpreting the results, thereby improving the overall organization of the manuscript.
Comment 7. While the enzymatic system is well thought out, I noticed that there is no mention of how the product of interest will be recovered from the mixture. Although I understand that developing a detailed downstream processing (DSP) methodology may not be necessary, it would be beneficial to at least outline potential purification strategies for the target compound(s). Given the complexity of the one-pot system, which contains enzyme buffer and other components, recovery could present significant challenges that might influence the overall feasibility of your work. Including this information would enhance the manuscript by providing a more complete picture of the practical application.
Response 7: We appreciate your insightful comments. Firstly, the development of enzyme systems for the biosynthesis of arabinosides presents significant challenges. While nucleosides have been successfully produced through fermentation, the market price of arabinosides is at least 10 times higher than that of nucleosides. Therefore, developing a biological conversion method for nucleosides to arabinosides holds substantial research and practical value.
Regarding the recovery of the target product from the one-pot system, the challenges are relatively manageable due to the distinct solubility differences between arabinosides, nucleosides, and bases in water. To address this, we typically employ strategies such as adjusting pH and temperature to precipitate the target product or remove impurities. For example, adenosine is nearly insoluble in water, while adenosine is highly soluble, and guanine dissolves in ammonia or potassium hydroxide solutions. In our one-pot conversion system from adenosine to ara-A, we induce precipitation of ara-A by adjusting the pH to alkaline, followed by heating and natural cooling to recover the product.
Although we recognize that a detailed downstream processing (DSP) methodology may not be necessary at this stage, we believe these preliminary purification strategies demonstrate the feasibility of product recovery within the context of our system.
Comment 8. The metrics discussed in the study should include space-time yields, as this measurement is critical for evaluating the efficiency of the enzymatic system. Comparing space-time yields with those from other studies or enzyme systems would provide valuable context for your findings and help demonstrate the potential advantages of your approach. This addition could strengthen the overall impact of the manuscript by highlighting its contributions to the field.
Response 8: We appreciate your valuable comments. Space-time yield data have been included in Figure 4 and discussed in the revised manuscript. In comparison with other studies, it is important to note that most studies calculate product yield relative to the substrate. Additionally, different catalytic methods often utilize different substrates, making direct comparisons challenging at this stage. Nevertheless, we believe that presenting our space-time yield provides useful context and demonstrates the efficiency of our enzymatic system.
Comment 9. Assessing how the enzyme activity declines under the given reaction conditions would provide valuable information about the system's stability and help identify factors that may contribute to enzyme degradation. This data could be crucial for understanding the long-term viability of the enzymatic process and optimizing reaction conditions to enhance enzyme longevity.
Response 9: We appreciate your insightful comments. Given the short reaction time, we assessed enzyme stability over a 2-hour period. The results indicated that enzyme activity remained essentially unchanged (data not shown). Although this preliminary data suggests stability under the current conditions, we recognize the importance of further studies to evaluate enzyme degradation over longer reaction periods and under varying conditions to optimize the system for long-term viability.
Comment 10. Comments on the Quality of English Language
I noticed that the term "thermodynamic dynamics" appears in line 228. It’s a bit redundant since thermodynamics already encompasses dynamic processes related to energy changes. Simplifying this to just "thermodynamics" would enhance clarity. Additionally, I've come across similar instances throughout the text. It might be beneficial to have the paper reviewed by a native English speaker or a professional editing service to improve the overall language quality. This could help strengthen the manuscript and ensure your findings are communicated as effectively as possible.
Response 10: We appreciate your thoughtful comments. We have revised the manuscript to eliminate redundancies, such as the term "thermodynamic dynamics," which has been simplified to "thermodynamics" for clarity. Additionally, we have carefully reviewed the language throughout the manuscript to improve overall clarity and coherence. We are also considering a final review by a native English speaker or professional editing service to ensure the manuscript meets the highest language standard.
Reviewer 2 Report
Comments and Suggestions for Authors
This article by Liu et al reports an investigation into the use of enzymes to facilitate the transformation of nucleosides into N-aryl-beta-D-arabinofuranosylamine, I found this manuscript difficult to read as the authors have routinely used incorrect nomenclature, left out anomeric stereochemistry, and mixed up singular and plural in their manuscript.
Although the authors have performed an interesting study, I found it difficult, in places, to follow the authors' arguments. That is, this manuscript needs revision before it could be consider for publication in Molecules.
I have detailed my comments below.
1) Throughout the manuscript the authors keep referring to their target compounds as arabinoside (singular!) but they should be called N-aryl-beta-D-arabinofuranosylamine or arabinofuranosyl nucleotide analogues. For example, the title of the manuscript must be changed to include "N-aryl-beta-D-arabinofuranosylamines" or "arabino-configured nucleoside analogues" (plural). That is, the use of arabinoside is confusing as it can refer to pyran (6-membered rings) or furan (5-membered rings) structures, D- or L- absolute configuration, alpha and beta (anomeric stereochemistry), and to O-, N-, or S-glycosides.
2) Scheme 1: why is there no stereochemistry shown on any of the anomeric carbons?
3) Scheme 2: part A "arabifuranyl" needs correcting. Anomeric stereochemistry is again missing.
4) Scheme 2: for these syntheses no anomeric ratios or yields are given so it's hard for the reader to evaluate whether the present multi-enzyme system gives better yields, anomeric selectivity, and/or ease of use.
5) Line 37, the authors mention "fludarabine", but the structure is not shown. I'd suggest adding it into Scheme 1.
6) Scheme 3: the abbreviations NDT is not given in the manuscript. Also, the scheme should not be shown with unidirectional arrows, enzyme catalysis speeds up reactions, but does not change the equilibrium constant, which in the example of panel A will be determined by the identity of the two nucleobases, their concentrations, and their solubility.
7) Line 199, the configuration not conformation of the hydroxy group at C2 not C2'.
8) Line 212, it is not a phosphorylation reaction it is phosphorolysis.
9) I had trouble understanding the point that the authors were trying to make with their computed free energy profile given that it does not include the starting nucleoside and the desired N-aryl-arabinosylamine product. The following sentence on line 224 is incorrect: "The Gibbs free energy change (ΔG°) of the overall pathway from R1P to A1P was analyzed to be -2.3 ± 2.4 kJ mol−1 (Figure 1B), and it is thermodynamic favorable to the synthesis of purine nucleoside and purine arabinoside from pentose 1-phosphate and nucleobase", given that the equilibrium between R1P and A1P is irrelevant to the overall equilibrium constant, which for some reason is shown in the SI and not in the paper. Also, insufficient details are given for the computation, what is the concentration of nucleobase. phosphate, and carbohydrate used in the computation? Also, are the computations done under the same conditions as the experiments?
10) Figure 4: The units for the y-axes (panel C) should include time, that is, concentration formed in unit time (rate), the time for incubation should be mentioned in the Figure legend.
Some required changes include:
1) Line 45; change "2,3,5-O-benzyl-D-arabinofuranosyl chloride" to "2,3,5-tri-O(ital)-benzyl-D(small cap)-arabinofuranosyl chloride"
2) Line 30; change "Arabinose nucleosides, also known as arabinoside, form…" to "Arabinose nucleosides, also known as arabinosylamines, form …".
3) Scheme 3, 6PGDH is an abbreviation that is not defined in this paper. The authors need to check thoroughly that ALL abbreviations are defined. If an abbreviation only appears once then just use the full name.
4) Line 237, phosphorylation and in numerous other places.
5) Line 296, rate constants use k not K.
Comments on the Quality of English LanguageNeed to check plural/singular in the MS.
Author Response
Comment 1. Throughout the manuscript the authors keep referring to their target compounds as arabinoside (singular!) but they should be called N-aryl-beta-D-arabinofuranosylamine or arabinofuranosyl nucleotide analogues. For example, the title of the manuscript must be changed to include "N-aryl-beta-D-arabinofuranosylamines" or "arabino-configured nucleoside analogues" (plural). That is, the use of arabinoside is confusing as it can refer to pyran (6-membered rings) or furan (5-membered rings) structures, D- or L- absolute configuration, alpha and beta (anomeric stereochemistry), and to O-, N-, or S-glycosides.
Response 1: Thank you for your valuable comments. We have revised the manuscript, including the title, to use the terms "N-aryl-beta-D-arabinofuranosylamine" and "arabino-configured nucleoside analogues" instead of "arabinoside" to eliminate any potential confusion regarding the compound's structure and configuration.
Comment 2. Scheme 1: why is there no stereochemistry shown on any of the anomeric carbons?
Response 2: We appreciate your insightful comment. The primary focus of the present multi-enzyme system is the sugar isomerization. However, to address your concern, the stereochemistry of the anomeric carbon on the ribosyl moiety has been included in the revised manuscript for clarity.
Comment 3. Scheme 2: part A "arabifuranyl" needs correcting. Anomeric stereochemistry is again missing.
Response 3: Thank you for your comment. "Arabifuranyl" has been corrected, and the anomeric stereochemistry has been included in the revised version of Scheme 2.
Comment 4. Scheme 2: for these syntheses no anomeric ratios or yields are given so it's hard for the reader to evaluate whether the present multi-enzyme system gives better yields, anomeric selectivity, and/or ease of use.
Response 4: We appreciate your thoughtful comments. We agree that including the anomeric ratios and yields would provide a clearer comparison of the performance of our multi-enzyme system. While chemical synthesis yields are generally high, which explains why arabinoside nucleosides are predominantly synthesized chemically, the enzymatic method holds significant potential for future applications. In the revised manuscript, we will highlight that, although direct yield comparisons are complex, enzymatic catalysis offers advantages in terms of selectivity and scalability.
Comment 5. Line 37, the authors mention "fludarabine", but the structure is not shown. I'd suggest adding it into Scheme 1.
Response 5: Thank you for your comment. The structure of "fludarabine" has been added to Scheme 1 in the revised manuscript.
Comment 6. Scheme 3: the abbreviations NDT is not given in the manuscript. Also, the scheme should not be shown with unidirectional arrows, enzyme catalysis speeds up reactions, but does not change the equilibrium constant, which in the example of panel A will be determined by the identity of the two nucleobases, their concentrations, and their solubility.
Response 6: We appreciate your insightful comments. The abbreviation "NDT" has been clarified and listed at the end of the title in Scheme 3. Additionally, a list of all abbreviations has been included at the end of the revised manuscript for better clarity.
Regarding the use of unidirectional arrows, we acknowledge that enzyme catalysis typically speeds up reactions without altering the equilibrium constant, which is determined by the identity, concentration, and solubility of the nucleobases. The unidirectional arrows in panel A were initially used to indicate that the reaction can proceed under specific conditions. However, to better represent the reversible nature of the enzymatic reactions, we have now updated the diagram with bidirectional arrows.
Comment 7. Line 199, the configuration not conformation of the hydroxy group at C2 not C2'
Response 7: Thank you for your comments. The term has been revised to "configuration" of the hydroxy group at C2, rather than "conformation," in the revised manuscript.
Comment 8. Line 212, it is not a phosphorylation reaction it is phosphorolysis.
Response 8: We appreciate your insightful comments. The term "phosphorylation" has been corrected to "phosphorolysis" in the revised manuscript.
Comment 9. I had trouble understanding the point that the authors were trying to make with their computed free energy profile given that it does not include the starting nucleoside and the desired N-aryl-arabinosylamine product. The following sentence on line 224 is incorrect: "The Gibbs free energy change (ΔG°) of the overall pathway from R1P to A1P was analyzed to be -2.3 ± 2.4 kJ mol−1 (Figure 1B), and it is thermodynamic favorable to the synthesis of purine nucleoside and purine arabinoside from pentose 1-phosphate and nucleobase", given that the equilibrium between R1P and A1P is irrelevant to the overall equilibrium constant, which for some reason is shown in the SI and not in the paper. Also, insufficient details are given for the computation, what is the concentration of nucleobase. phosphate, and carbohydrate used in the computation? Also, are the computations done under the same conditions as the experiments?
Response 9: Thank you for valuable comments. We have revised the analysis of the ΔG° change, and the ΔG° of the overall pathway from R1P to A1P is now shown in the Supporting Information (SI). The primary focus of the reaction in this multi-step biocatalytic cascade is sugar isomerization. Additionally, the overall pathway from R1P to A1P is consistent across the production of different arabinosides, which is why we focused on analyzing the ΔG° of this pathway. For the computation, standard concentrations of 1 M for the nucleobase, phosphate, and carbohydrate were assumed. The computations were performed under the same conditions as used in the experiments.
Comment 10. Figure 4: The units for the y-axes (panel C) should include time, that is, concentration formed in unit time (rate), the time for incubation should be mentioned in the Figure legend.
Response 10: We appreciate your insightful comment. The units for the y-axes in panel C have been revised to include time, reflecting the concentration formed per unit time (rate). Additionally, the incubation time has been mentioned in the Methods section (Section 2.6, "Production of arabinosides from nucleosides").
Comment 11. Some required changes include:
1) Line 45; change "2,3,5-O-benzyl-D-arabinofuranosyl chloride" to "2,3,5-tri-O(ital)-benzyl-D(small cap)-arabinofuranosyl chloride"
2) Line 30; change "Arabinose nucleosides, also known as arabinoside, form…" to "Arabinose nucleosides, also known as arabinosylamines, form …".
3) Scheme 3, 6PGDH is an abbreviation that is not defined in this paper. The authors need to check thoroughly that ALL abbreviations are defined. If an abbreviation only appears once then just use the full name.
4) Line 237, phosphorylation and in numerous other places.
5) Line 296, rate constants use k not K.
Response 11: Thank you for your comments. These errors have since been corrected in the revised manuscript.
Round 2
Reviewer 1 Report
Comments and Suggestions for Authors
The authors have incorporated changes to the manuscript according to the initial review report. The change of title makes sense to attract to a greater audience.
The manuscript can be now accepted as such.